# Cooperation between SS18-SSX1 and miR-214 in Synovial Sarcoma Development and Progression

**DOI:** 10.3390/cancers12020324

**Published:** 2020-01-30

**Authors:** Miwa Tanaka, Mizuki Homme, Yukari Yamazaki, Keisuke Ae, Seiichi Matsumoto, Subbaya Subramanian, Takuro Nakamura

**Affiliations:** 1Division of Carcinogenesis, The Cancer Institute, Japanese Foundation for Cancer Research, Tokyo 135-8550, Japan; miwa.tanaka@jfcr.or.jp (M.T.); mizuki.homme@jfcr.or.jp (M.H.); yukari.yamazaki@jfcr.or.jp (Y.Y.); 2Department of Orthopedic Oncology, The Cancer Institute Hospital, Japanese Foundation for Cancer Research, Tokyo 135-8550, Japan; keisuke.ae@jfcr.or.jp (K.A.); smatsumoto@jfcr.or.jp (S.M.); 3Department of Surgery, University of Minnesota, Minneapolis, MN 55455, USA; subree@umn.edu

**Keywords:** synovial sarcoma, SS18-SSX1, mouse model, insertional mutagenesis, miR-214

## Abstract

SS18-SSX fusion proteins play a central role in synovial sarcoma development, although, the genetic network and mechanisms of synovial sarcomagenesis remain unknown. We established a new ex vivo synovial sarcoma mouse model through retroviral-mediated gene transfer of *SS18-SSX1* into mouse embryonic mesenchymal cells followed by subcutaneous transplantation into nude mice. This approach successfully induced subcutaneous tumors in 100% recipients, showing invasive proliferation of short spindle tumor cells with occasional biphasic appearance. Cytokeratin expression was observed in epithelial components in tumors and expression of TLE1 and BCL2 was also shown. Gene expression profiling indicated SWI/SNF pathway modulation by *SS18-SSX1* introduction into mesenchymal cells and *Tle1* and *Atf2* upregulation in tumors. These findings indicate that the model exhibits phenotypes typical of human synovial sarcoma. Retroviral tagging of the tumor identified 15 common retroviral integration sites within the *Dnm3* locus as the most frequent in 30 mouse synovial sarcomas. *miR-199a2* and *miR-214* upregulation within the *Dnm3* locus was observed. *SS18-SSX1* and *miR-214* cointroduction accelerated sarcoma onset, indicating that *miR-214* is a cooperative oncomiR in synovial sarcomagenesis. *miR-214* functions in a cell non-autonomous manner, promoting cytokine gene expression (e.g., *Cxcl15/IL8*). Our results emphasize the role of *miR-214* in tumor development and disease progression.

## 1. Introduction

Synovial sarcoma is a malignant mesenchymal neoplasm characterized by *SS18-SSX* gene fusions associated with t(X;18) chromosome translocation, characteristic biphasic morphologies consisting of both mesenchymal and epithelial components, highly invasive growth, and poor prognosis [1,2,3]. *SS18-SSX1/SSX2/SSX4* fusions were observed in most synovial sarcoma [4]. The SS18-SSX fusion proteins show pleiotropic functions that promote oncogenesis [5]. SS18-SSX1 competes with wild type SS18, a component of the SWI/SNF complex, hijacking the complex, inducing aberrant expression of *Sox2* and perturbating the interaction between polycomb and SWI/SNF complexes [6,7]. SS18-SSX1 also interacts with the transcriptional co-repressor TLE1 and recruits it to ATF2, resulting in abnormal repression of ATF2 target genes such as *EGR1* [8]. In addition, SWI/SNF disruption by SS18-SSX fusion proteins induces activation of WNT/β-catenin signaling [9]. Another study showed that KDM2B and polycomb repressive complex 1 recruits SS18-SSX1 proteins and the SWI/SNF complex to unmethylated CpG islands, leading to aberrant activation of gene expression [10]. Although *PTEN*, *CTNNB1*, and *APC* mutations have been reported in synovial sarcoma, the frequency of secondary mutations in this disease is not very high [11]. Moreover, the cell-of-origin of synovial sarcoma remains unclear [1], indicating that the genetic/epigenetic events that cooperate with SS18-SSX should be clarified.

A genetically engineered mouse model of human synovial sarcoma has been previously created by introducing a conditional allele of *SS18-SSX2* into the *ROSA26* locus, after which the knock-in mice were by mated with *Myf5*-Cre transgenic mice. As a result, the biphasic synovial sarcoma phenotype was well recapitulated [12], indicating that *SS18-SSX2* expression efficiently induces synovial sarcoma in a specific cellular lineage. We have previously generated mouse models of fusion gene-associated sarcoma by introducing *EWS-FLI1*, *ASPSCR1-TFE3*, or *CIC-DUX4* into mouse embryonic mesenchymal cells (eMCs) [13,14,15]. In these models, induced tumors can recapitulate phenotypes of human counterparts and are therefore useful as platforms for exploring novel disease mechanisms and testing the effects of new drugs.

In this study, we generated a new mouse model of human synovial sarcoma by using the ex vivo gene manipulation technology and mouse embryonic mesenchymal cells. Moreover, the retrovirus-mediated gene transfer procedure allowed us to perform retroviral tagging and to identify novel cooperative genes for *SS18-SSX1* in synovial sarcoma development. We showed that expression of microRNA *miR-214* accelerates the onset of synovial sarcoma. Given that overexpression of *miR-214* in human synovial sarcoma has been previously reported [16], our present results shed light on the mechanisms of synovial sarcoma development and progression.

## 2. Results

### 2.1. Generation and Characterization of the Ex Vivo Model of Synovial Sarcoma

In previous studies, we generated mouse models of human sarcoma expressing fusion genes such as Ewing sarcoma, alveolar soft part sarcoma, and CIC-DUX4 sarcoma [13,14,15]. Our work emphasizes the advantages of using mouse eMCs to concentrate a cell-of-origin for fusion gene-associated sarcomas. In this context, we retrovirally introduced human *SS18-SSX1* into murine eMCs derived from embryo extremities on dpc 18.5. Transduced eMCs were then subcutaneously injected into nude mice. Recipient mice developed a subcutaneous mass with 100% penetrance and a mean latency of 26.6 weeks (Figure 1A). These tumors were slow growing and invaded surrounding tissues but without showing distant metastasis.

Histological analysis showed that *SS18-SSX1*-expressing tumors were composed of a diffuse proliferation of short spindle cells with or without tubulus-like epithelial structures, equivalent to human biphasic or monophasic synovial sarcoma, respectively (Figure 1B). Of the 59 sarcomas, 44 were monophasic and 15 were biphasic. Sarcoma cells showed diffuse infiltration into the surrounding tissues such as skeletal muscles where intra-fascicular invasion was evident (Figure 1C). Expression of FLAG-tagged SS18-SSX1 fusion proteins was confirmed by immunoblotting (Figure 1D), and nuclear accumulation of fusion proteins was observed both in spindle and epithelial elements (Figure 1E). Immunostaining showed expression of cytokeratin (AE1/AE3) not only in epithelial-like cells but also in spindle cells in 19 out of 20 tumors. Enhanced expression of BCL2 and TLE1 was consistently observed in 20 out of 20 tumors with observations in the human counterpart (Figure 1F) [17,18]. These tumors were transplantable into nude mice and Balb/c mice with similar morphological characteristics (Figure 1G), indicating that the tumors were not transient overgrowths of eMCs. Together, these results show that introduction of *SS18-SSX1* into eMCs can induce sarcomas which recapitulated the phenotypes of human synovial sarcoma, such as typical biphasic morphologies and expression of synovial sarcoma-specific biomarkers.

### 2.2. Gene Expression Profile of Mouse Synovial Sarcoma

Microarray analyses were carried out to examine the gene expression profile of mouse synovial sarcomas and mouse eMCs with or without introduction of SS18-SSX1 at 48 h after introduction. Gene set enrichment analysis (GSEA) using gene sets for eMCs showed enrichment of the SNF5 gene set containing genes upregulated in mouse embryonic fibroblasts with *Snf5* knockout [19] in the *SS18-SSX1*-expressing cohort (Figure 2A and Appendix A), and the changes were detected 48 h after *SS18-SSX1* introduction. Since disruption of the SNF/SWI complex by SS18-SSX1 was observed in human synovial sarcoma cell lines [7], the present result suggests that affection of the pathway occurred at the early stage of synovial sarcoma development. Representative downstream genes in the SNF5 pathway, *Sox2* and *ApoD,* were significantly upregulated in eMCs expressing *SS18-SSX1* and also in mouse synovial sarcomas (Figure 2B). Expression profiles of mouse synovial sarcomas were also analyzed, and enriched genes within Cyclin D1, PTEN, and β-catenin pathways were also found to be upregulated (Figure 2C and Appendix A). Upregulation of *ATF2* and *TLE1* is important in the development of human synovial sarcoma and recruitment of TLE1 to the ATF2 binding locus downregulates expression of the tumor suppressor *EGR1* [8,20,21]. In accordance with this, expression of *Atf2* and *Tle1* was increased, while that of *Egr1* was suppressed in mouse synovial sarcoma (Figure 2D). These changes were not observed in eMCs expressing *SYT-SSX1*, suggesting that the expression modulation of the Atf2/Tle1/Egr1 axis might be achieved at a later stage. However, *Igf1*, which is downstream of the Cyclin D1 pathway, was upregulated both in tumors and eMCs expressing *SS18-SSX1*, which might be important for drug resistance in synovial sarcoma treatment [22]. Overall comparison of differentially expressed genes between *SS18-SSX1*-expressing eMCs and tumors showed that 3155 and 485 genes were upregulated in synovial sarcoma versus eMCs with an empty vector; fold change > 2.0, and in *SS18-SSX1*-expressing eMCs versus eMCs with an empty vector; fold change > 2.0, (Figure 2E and Appendix A). Among these genes commonly upregulated genes in *SS18-SSX1*-expressing eMCs and tumors, such as *Sox2* and *ApoD,* might be downstream targets of the SS18-SSX1 and SWI/SNF axis, and their expression is preserved during oncogenesis, whereas tumor-specific upregulated genes such as *Atf2* and *Tle1* might require other factors for their expression at the later stage.

### 2.3. Retroviral Tagging Identifies Candidate Cooperative Genes for SS18-SSX1

In the present mouse synovial sarcoma model, *SS18-SSX1* was introduced using a murine myeloid leukemia virus-based retroviral vector, which enabled aberrations of genes cooperating with *SS18-SSX1* via retroviral integrations [23,24,25]. Southern blotting showed that tumors showed integrations of multiple retroviral copies (Appendix A). To identify such cooperative genes, 269 retroviral integration sites were isolated and 15 were common in 30 synovial sarcomas (Figure 3A, Table 1 and Appendix A). The most frequent integration site was the *Dnm3* locus on chromosome 1 in which three tumors, SS18, SS19, and SS23, showed retroviral insertions (Figure 3B). These integrations were observed in introns of *Dnm3* without upregulating its expression (Appendix A). There is a *Dnm3os* long non-coding RNA in intron 14 of *Dnm3* with opposite transcriptional direction. In addition, *Dnm3os* contains two microRNAs, *miR-199a2* and *miR-214* (Figure 3B) [26].

Expression of *miR-199a2* and *miR-214* were significantly upregulated in synovial sarcomas (SS18, SS19, and SS23) with retroviral integrations, although the upregulation of *miR-214* in SS23 was moderate (Figure 3C). It is possible that the retroviral integration in the SS23 locus might be present only in a minor subpopulation of the tumor. Moreover, *miR-199a2* upregulation was observed only in one tumor (SS19) with retroviral integrations, suggesting that *miR-199a2* might not be a target of the integration as was also indicated by the co-expression experiment (see below). *miR-214* expression was also upregulated in human synovial sarcoma as reported previously [16], and its expression was the highest in sarcomas of six types (Appendix A). Increased *Dnm3os* expression was also observed both in human and mouse synovial sarcomas compared with Ewing sarcoma, alveolar soft part sarcoma, or rhabdomyosarcoma (Appendix A), suggesting the importance of the locus in synovial sarcoma development.

### 2.4. miR-214 Cooperates with SS18-SSX1 in Synovial Sarcoma Development

To examine the cooperativity between *SS18-SSX1* and *miR-199a2* or *miR-214*, *SS18-SSX1*, either microRNA sequences were coexpressed in eMCs and subjected to sarcoma induction experiments (Figure 4A). Coexpression of *miR-214* but not *miR-199a2* significantly accelerated *SS18-SSX1*-induced synovial sarcoma development with 14.4 and 29.2 weeks of mean latency, respectively (Figure 4B). Expression of *Fgfr1* and *Nedd9*, also located near common integration sites, did not accelerate synovial sarcoma development (Appendix A). *miR-214* expression was compared between *SS18-SSX1/miR-214* co-transduced tumors and *SS18-SSX1* only tumors, and a significant inverse correlation was observed between *miR-214* expression and the sarcoma latency (Figure 4C). *miR-214*-positive sarcomas showed similar morphological characteristics to *SS18-SSX1* only sarcomas, namely monophasic or biphasic patterns composed of short spindle tumor cells (Figure 4D), and there was no significant difference in SS18-SSX1 protein expression (Figure 4E). In situ hybridization showed *miR-214* expression in both the nucleus and cytoplasm of *SS18-SSX1-* and *miR-214*-coexpressing tumor cells (Figure 4F), suggesting that *miR-214* mainly functions in tumor cells. Subsequently, we found that expression of *miR-214* was correlated with a worse prognosis in human synovial sarcoma cases (Figure 4G).

### 2.5. Effects of miR-214 Expression on Synovial Sarcoma

Gene expression profiles were compared between synovial sarcomas expressing *SS18-SSX1* (*n* = 12) and *SS18-SSX1* with *miR-214* (*n* = 13). The principal component analysis and unsupervised clustering showed that both cohorts had distinct expression profiles in both subtypes with a minor exception (Figure 5A,B). We selected 168 genes that were downregulated following introduction of *miR-214* in two different synovial sarcoma cell lines, SS40 and SS60, and 53 genes that were potential *miR-214* targets were further selected by a database search (miRbase; http://www.mirbase.org and TargetScan; http://www.targetscan.org/vert_72/). Among these 53 genes, expression of 11 genes (*Timp2*, *Ezh1*, *Foxo4*, *Nomo1*, *Pten*, *Tpp1*, *Plagl2*, *Chfr*, *Fbxo3*, *Lats2*, and *Cbl*) that are predicted targets of *miR-214* were further analyzed because of their known involvement in cancer (Figure 5C and Appendix A). RT-qPCR showed downregulation of *Timp2*, *Ezh1,* and *Pten* and immunoblotting showed that the effect of *miR-214* on these genes was rather limited (Figure 5C,D). Nevertheless, exogenous introduction of *miR-214* or *Dnm3os* into synovial sarcomas and silencing of *miR-214* by *anti-miR-214* did not induce growth promotion, migration, or invasion (Appendix A), suggesting that *miR-214* might not function in a cell intrinsic manner.

Ingenuity pathway analysis (IPA) of differentially expressed genes between synovial sarcomas with and without *miR-214* expression showed an enrichment for inflammatory response genes (Figure 5E and Appendix A). In addition, a series of cytokine and chemokine genes were upregulated in synovial sarcomas expressing *miR-214* (Figure 5F and Table 2). Cytokine/chemokine expression was compared between human synovial sarcoma and other sarcomas such as osteosarcoma, Ewing sarcoma, alveolar soft part sarcoma, and rhabdomyosarcoma using publicly available databases (Table 3). The mouse *Cxcl15/Il8*, a counterpart of human *CXCL8/IL8,* was upregulated in both human synovial sarcoma and mouse synovial sarcoma with *miR-214* expression. Microarray analyses of mRNA and microRNA expression in mouse synovial sarcoma showed a strong positive correlation (r = 0.77, *p* < 0.01) between *Cxcl15* and *miR-214* expression (Figure 5G). Cxcl15 transcript and protein levels in *miR-214*-positive synovial sarcomas was confirmed by RT-qPCR and ELISA, respectively (Figure 5H,I). Cxcl15/IL-8 induces migration of macrophages and granulocytes. However, the number of CD163-positive macrophages and naphthol AS-D chloroacetate esterase-positive granulocytes were not different between synovial sarcoma with and without *miR-214* expression (Appendix A).

## 3. Discussion

Our mouse model of synovial sarcoma showed faithful reproduction of the phenotype of its human counterpart using eMCs as the cell-of-origin. Biphasic morphology consisting of both mesenchymal and epithelial components, a hallmark of synovial sarcoma pathology, was observed in 25% of all tumors. The incidence of the biphasic subtype was low but cytokeratin-positive tumor cells were frequently observed in the monophasic mesenchymal subtype, indicating that there is transition between the two components. Previous studies showed that myoblast-specific expression of *SS18-SSX2* induced synovial sarcoma phenotypes in mouse [12], suggesting that myogenic progenitors might be one of the cell-of-origin of synovial sarcoma. Although our current model did not contribute to the topological information for the cell-of-origin, our results using embryonic mesenchymal cells suggest an association between some developmental abnormalities and synovial sarcomagenesis.

The micro RNAs *miR-214* and *miR-199a-2* are embedded as parts of lncRNA *Dnm3os* within the *Dnm3* locus [24]. The locus is well conserved during evolution and common target genes of *miR-214* have been reported among human, mouse, and zebrafish [27]. Mesenchyme-specific expression of *Dnm3os* has been reported and *Dnm3os* knockout mice show severe abnormalities in musculoskeletal development [26,27,28,29]. The importance of *miR-214* in mesenchymal tissue and the high level of expression of *Dnm3os* suggests the presence of *miR-214*-expressing clusters in embryonic mesenchymal cells. Although upregulation of *miR-214* in our mouse model was achieved by insertions of retroviral sequences that act as an enhancer [23], the mechanism of *miR-214* and *DNM30S* upregulation in human synovial sarcoma remains unclear. In addition, *miR-214* and *miR-199a* upregulation was observed in several mouse synovial sarcomas without retroviral integrations. Expression of *Dnm3os* suggests epigenetic regulation of the locus in the developing mesenchymal tissues, and further studies are needed to clarify the mechanism [26]. A previous study has reported that the *Dnm3os* locus is regulated by TWIST1, which promotes cancer stemness, inflammation, and proliferation of human ovarian cancer cells [30]. In this case, upregulation of *miR-199a* induces pro-inflammatory environment via NF-κB activation, suggesting that different mechanisms exist between ovarian cancer and synovial sarcoma. Although the exact cell-of-origin of synovial sarcoma remains to be clarified, the results indicate important functions of *miR-214* in the growth and differentiation of mesenchymal cells. The cooperative effect of *miR-214* for *SS18-SSX1* in the mouse model and the association between a high expression of *miR-214* with a worse prognosis in human synovial sarcoma cases strongly suggest a role of *miR-214* in the aggressive phenotype (Figure 6).

Hirata et al. have recently identified gene fusions between *CTDSP1/2* and *DNM3OS* in a subset of dedifferentiated liposarcomas [31]. The fusions induce upregulation of both *DNM3OS* and *miR-214*, as seen in the murine synovial sarcoma with retroviral integration at the *Dnm3os* locus. Increased expression of *DNM3OS* and *miR-214* was associated with worse prognosis in dedifferentiated liposarcoma, indicating common functions for *miR-214* in immature and aggressive phenotypes of soft part sarcomas.

Despite upregulation of *miR-214* in both mouse and human synovial sarcoma [16], exogenous introduction of *miR-214* into synovial sarcoma cells did not promote growth, cell migration, or invasion. Tumor suppressor *Pten* has been reported to be a target of *miR-214* in human cancer [32], and its expression was indeed reduced by *miR-214*. In our study, downregulation of *Pten* did not show a remarkable growth promoting effect (Appendix A), although a previous study showed that *Pten* knockout in the *SS18-SSX1-* and *SS18-SSX2*-induced synovial sarcoma mouse models promoted invasion and metastasis [33].

*miR-214* overexpression in synovial sarcoma cells did not promote cell growth, migration, nor invasion abilities in vitro, suggesting that it might promote tumor development in a cell non-autonomous manner [34,35]. Gene expression profiling revealed upregulation of inflammatory cytokine genes in synovial sarcoma with *miR-214* overexpression. Among upregulated cytokine genes *Cxcl15* which encodes for IL-8, the mouse homolog of human CXCL8/IL-8, was upregulated both in mouse and human *miR-214*-overexpressing synovial sarcoma. IL-8 induces migration and activation of granulocytes and macrophages [36,37]. Although *miR-214*-positive sarcoma failed to show a significant increase in granulocyte or macrophage infiltration in the tumor tissue, increased cytokines might activate intracellular signaling of inflammatory cells to secrete tumor growth promoting factors. Given the high invasive potency of synovial sarcoma cells towards the surrounding tissue, it is possible that IL-8 contributes to the invasive activity of the tumor cells by interacting with the tumor microenvironment. Indeed, increased secretion of IL-8 secretions has been reported in various human malignancies, and IL-8 is involved in more aggressive phenotypes [38,39,40,41]. Functional modulation of tumor-associated neutrophils by IL-8 has also been described as a poor prognostic factor for malignancies [39,42]. As the tumor promoting effect of *Dnm3os* via cancer-associated fibroblasts was also demonstrated in esophageal cancer [43], dynamic modulation of the tumor microenvironment might also be achieved by the *Dnm3os/miR-214* axis. In our synovial sarcoma model, nude mice lacking a T-cell population were used given that primary tumors have a slow growth in immunocompetent mice. Nevertheless, significant acceleration of sarcoma development was achieved by *miR-214* expression, indicating that T-cell function is dispensable for *miR-214*-induced modulation of the tumor microenvironment.

The mechanisms by which *miR-214* upregulates the expression of cytokine genes remain to be clarified. Modulations of the adenosine A2A receptor and the TGF-β signaling by *miR-214* have been proposed to have potential roles in cytokine secretion and in the inflammatory response [44,45]. A previous study showed that long non-coding RNA *NEAT1* transactivates IL-8 by relocating SFPQ to the nuclear paraspeckle [46]. Similarly, *miR-214* might be involved in transactivation of cytokine genes. In conclusion, our study indicates the importance of cooperation between *SS18-SSX1* and *miR-214* expression in the development and malignant phenotypes of synovial sarcoma. Further studies are needed to clarify the mechanistic role of *miR-214* upregulation in synovial sarcoma progression.

## 4. Materials and Methods

### 4.1. Plasmid Construction

N-terminal FLAG-tagged *SS18-SSX1* was introduced into the pMYs-IRES-GFP vector. Full-length *SS18-SSX1* was cloned from a human synovial sarcoma case. To generate pMYs-SS18-SSX1-IRES-miR-214 and pMYs-SS18-SSX1-IRES-miR-199a2, a mouse 200 bp genomic DNA sequence containing *miR-214* or *miR-199a2* was introduced by replacing it with the GFP sequence, respectively.

### 4.2. Generation and Characterization of the Mouse Synovial Sarcoma Model

Mouse embryonic mesenchymal cells (eMCs) were prepared as previously described [14]. Briefly, 18.5 dpc embryos were obtained from pregnant BALB/c mice and mesenchymal cells were dissociated from limb soft parts. eMCs were transduced with a pMYs-SS18-SSX1 and 1 × 10^6^ cells were subcutaneously transplanted into nude mice within 48 h after preparation of eMCs. All animal experiments were approved by the animal care committee at the Japanese Foundation for Cancer Research under licenses 10-05-9 and 0604-3-13.

### 4.3. Human Sarcoma Specimens

Synovial sarcoma, rhabdomyosarcoma, myxoid liposarcoma, dermatofibrosarcoma protuberans Ewing sarcoma, and solitary fibrosarcoma surgical specimens were obtained from the adult patients at the Cancer Institute Hospital, Tokyo, Japan after they had provided written informed consent. The study was conducted in accordance with ethical guidelines and approved by Institutional Review Board at the Japanese Foundation for Cancer Research under license 2013-1155.

### 4.4. Histology and Immunohistochemistry

Formaldehyde-fixed, paraffin-embedded tumor tissues were stained with hematoxylin and eosin (H&E) using standard techniques. Immunohistochemistry was performed using primary antibodies in conjunction with the Histofine Simple Stain kit (Nichirei, Tokyo, Japan). The following primary antibodies were used: Anti-FLAG M2 (Sigma, Burlington, MA, USA), anti-AE1/AE3 (DAKO, Santa Clara, CA, USA), anti-BCL2 (Santa Cruz Biotechnology, Dallas, TX, USA), anti-TLE1 (Santa Cruz Biotechnology), and anti-CD163 (Bioss Antibodies, Woburn, MA, USA).

### 4.5. Immunoblotting

Immunoblotting was performed using whole-cell lysates as previously described [47]. The following primary antibodies were used: Anti-FLAG (M2, Sigma), anti-α-tubulin (Sigma), anti-Pten (Cell Signaling Technology, Danvers, MA, USA), anti-Timp2 (Cell Signaling Technology), anti-Ezh1 (Santa Cruz), and anti-Gapdh (Hytest, Turku, Finland).

### 4.6. Microarray Analysis

GeneChip analysis was performed as previously described [14]. Comparison between 12 synovial sarcoma samples, four replicates of eSZ cells, and four replicates of normal human tissue mix consisting of skin, bone marrow, ovary, intestine, kidney, muscle, heart, liver, lung, and brain was achieved. Expression data were analyzed using GeneSpring ver. 14.9 (Agilent Technologies, Santa Clara, CA, USA). For pathway enrichment analysis, the GSEA-P 2.0 software was used [48]. Gene ontology analysis was performed using the Ingenuity Pathway Analysis (IPA) ver. 01-16. The microarray data sets are accessible through the NCBI Gene Expression Omnibus (GEO) database (http://www.ncbi.nlm.nih.gov/geo), with accession number GSE141251. Gene expression data of human sarcoma samples were obtained from the GEO database (GSE20196, GES87437, GSE12102, GSE32569, and GSE66533) [49,50,51,52,53].

### 4.7. Real-Time Quantitative Polymerase Chain Reaction (RT-qPCR)

Total RNA extraction, reverse transcription, and RNA quantification were performed according to previously described methods [12]. MicroRNA isolation was performed using the mirVana miRNA isolation kit (Thermo Fisher Scientific, Waltham, MA, USA). Quantification of microRNA was performed using the TaqMan MicroRNA assay kit (Thermo Fisher Scientific). Probes and primers for *pri-miR-214*, *miR-214_3p*, *miR-214_5p*, *pri-miR-199a-2*, *miR-199a-2_3p*, *miR-199a-2_5p*, *sno202*, and *u6* were purchased from Thermo Fisher Scientific. The sequences of the oligonucleotide primers used are shown in Appendix A.

### 4.8. Cloning of Retroviral Integration Sites

Retroviral integration sites were identified by the inverse-PCR method previously described [22,23]. Genomic DNA sequences flanking the retrovirus were cloned into the pGEM T-easy plasmid (Promega, Madison, WI, USA), sequenced, and mapped on mouse chromosome.

### 4.9. In Situ Hybridization

Expression of *miR-214* was detected in formaldehyde-fixed, paraffin-embedded sections of mouse synovial sarcoma using a Hsa-miR-214 probe (BioGenex, Fremont, CA, USA) and the Super Sensitive One-Step Polymer-HRP ISH Detection System (BioGenex) according to the manufacturer’s instructions. miR-U6 and miR-Scramble probes (BioGenex) were used as a positive and a negative control, respectively.

### 4.10. ELISA

Mouse synovial sarcoma tissue samples were analyzed using the Mouse Interleukin 8 (IL-8) ELISA kit (MyBioSource, San Diego, CA, USA) according to manufacturer’s instructions.

### 4.11. Statistical Analysis

All data are expressed as the mean ± SEM. Continuous distributions were compared with a two-tailed Student *t*-test. Survival analysis was performed using the Kaplan–Meier life table method, and survival between groups was compared with the log–rank test. The estimated hazard ratio (HR) and its two-sided 95% confidence interval (CI) were indicated. For significant test results in the miR-214 expression and overall survival analysis, the receiver operating characteristics (ROC) curve was generated to further assess the potential for utility as patient selection markers. Correlation between two groups was tested using Pearson’s correlation analysis.

## 5. Conclusions

Our ex vivo mouse model demonstrates both morphological and transcriptional characteristics of human synovial sarcoma. Upregulation of *miR-214* accelerates development of mouse synovial sarcoma via modulation of cytokine gene expression and tumor microenvironment, cooperating with *SS18-SSX1*. Given its association with a worse prognosis in human synovial sarcoma cases, *miR-214* can be used as a prognostic biomarker.

## Figures and Tables

**Figure 1 cancers-12-00324-f001:**
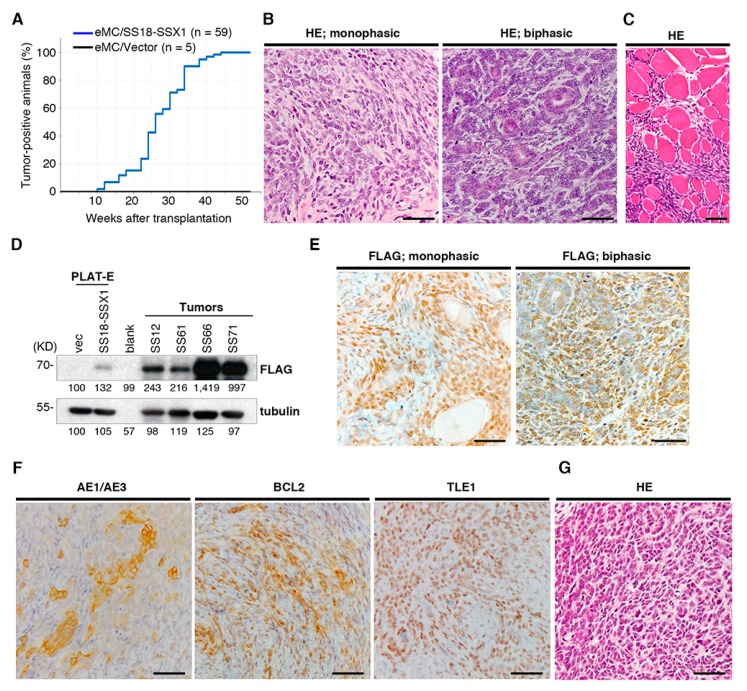
Generation of the mouse synovial sarcoma ex vivo model. (**A**) Cumulative incidences of synovial sarcoma. (**B**) Histology of mouse synovial sarcoma showing monophasic (**left**) and biphasic (**right**) patterns. (**C**) Infiltration of sarcoma cells into surrounding muscle tissue. (**D**) Immunoblotting of FLAG-SS18-SSX1 in tumor tissue. PLAT-E cells transduced with *SS18-SSX1* or empty vector are used as positive and negative controls, respectively. (**E**) Accumulations of SS18-SSX1 in the nucleus of tumor cells are evident in both monophasic and biphasic tumors by immunostaining with anti-FLAG. (**F**) Immunostaining with anti-AE1/AE3, anti-BCL2, and anti-TLE1. (**G**) Histology of the secondary transplanted tumor. Scale bars indicate 50 μm.

**Figure 2 cancers-12-00324-f002:**
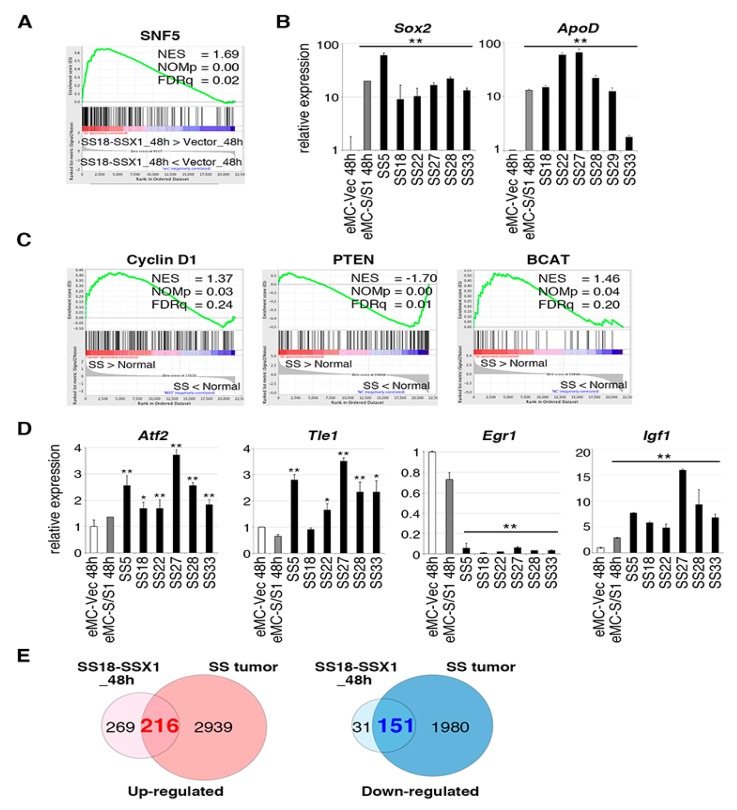
Gene expression profiling of mouse synovial sarcoma and embryonic mesenchymal cells (eMCs) expressing *SS18-SSX1*. (**A**) Gene set enrichment analysis (GSEA) of eMCs expressing *SS18-SSX1* versus eMCs with an empty vector shows enrichment of the SNF5 pathway. (**B**) Real-time quantitative RT-PCR for *Sox2* and *ApoD* in eMCs with or without *SS18-SSX1*, and synovial sarcoma tissues. Expression values were normalized to *Gapdh* expression. ** *p* < 0.01. (**C**) GSEA of mouse synovial sarcomas versus mouse normal tissue mix shows enrichment of the Cyclin D1, Phosphatase and Tensin Homolog (PTEN), and β-catenin pathways. (**D**) RT-qPCR for *Atf2*, *Tle1*, *Egr1*, and *Igf1* in eMCs with or without *SS18-SSX1*, and synovial sarcoma tissues. Expression values were normalized to *Gapdh* expression. * *p* < 0.05 and ** *p* < 0.01. (**E**) Venn diagrams for upregulated and downregulated genes in eMCs expressing SS18-SSX1 (versus eMCs with empty vector) and mouse synovial sarcomas (versus mouse normal tissue mix). Fold changes > 2.0, *p* < 0.05.

**Figure 3 cancers-12-00324-f003:**
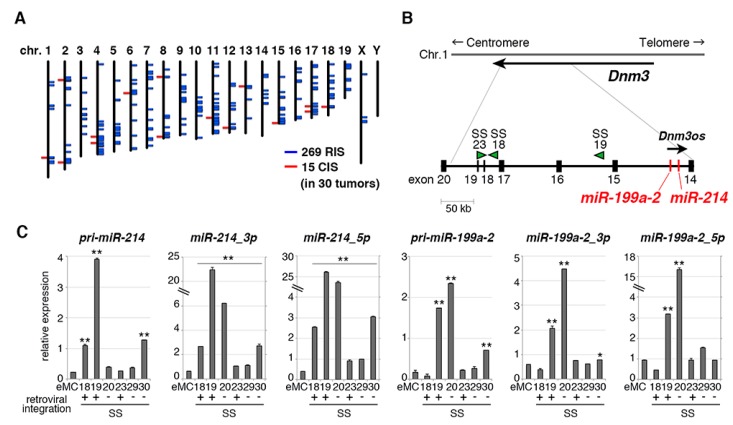
Identification of retroviral integration sites in mouse synovial sarcomas. (**A**) Distribution of 269 retroviral integration sites in 30 tumors on mouse chromosomes. Common integration sites are indicated with red lines, and single integration sites with blue lines. (**B**) Schematic of retroviral integrations at the *Dnm3* locus. Each integration is marked with arrowheads with the directions also indicated. Genetic positions of *Dnm3os*, *miR-214*, and *miR-199a-2* are indicated. (**C**) RT-qPCR for *pri-miR-214*, *miR-214_3p*, *miR-214_5p*, *pri-miR-199a-2*, *miR-199a-2_3p*, and *miR-199a-2_5p* in synovial sarcomas with or without retroviral integrations. Expression values were normalized to *snoRNA202* expression. * *p* < 0.05 and ** *p* < 0.01.

**Figure 4 cancers-12-00324-f004:**
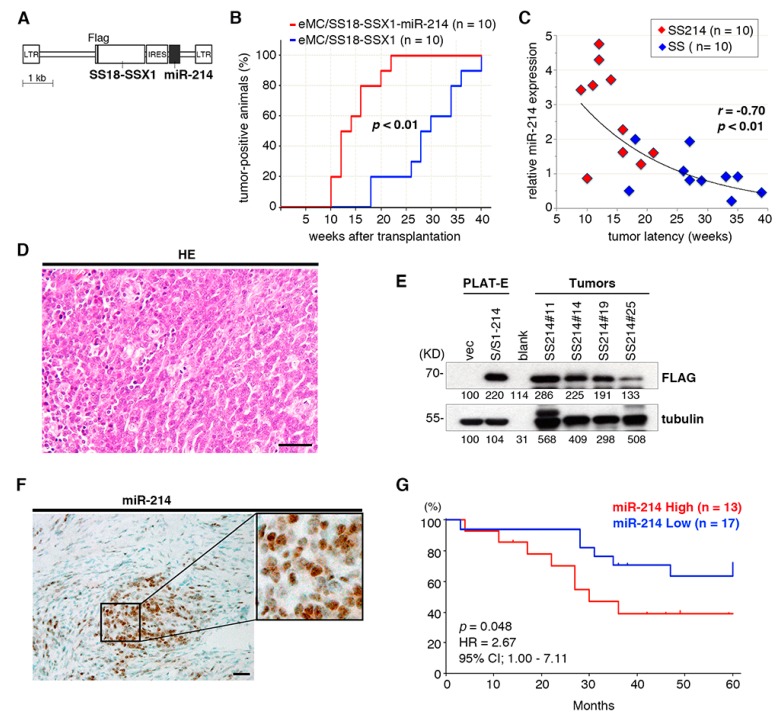
*miR-214* cooperates with *SS18-SSX1* in synovial sarcoma development. (**A**) Schematic of pMYs-SS18-SSX1-IRES-miR-214/miR-199a-2. (**B**) Cumulative incidences of synovial sarcoma with or without *miR-214*. Significance was examined by log-rank test. (**C**) Pearson’s correlation analysis shows inverse correlation between *miR-214* expression and disease latencies. (**D**) Histology of synovial sarcoma expressing both *SS18-SSX1* and *miR-214*. Scale bar indicates 50 μm. (**E**) Immunoblotting of FLAG-SS18-SSX1 in tumor tissue shows no significant difference in expression between sarcomas with and without *miR-214*. (**F**) In situ hybridization of *miR-214* shows sarcoma cell-specific expression of *miR-214.* Inset clarifies nuclear and cytoplasmic accumulation of *miR-214.* Scale bar indicates 50 μm. (**G**) Kaplan–Meier survival curves show that increased expression of *miR-214* is associated with poor prognosis in human synovial sarcoma cases.

**Figure 5 cancers-12-00324-f005:**
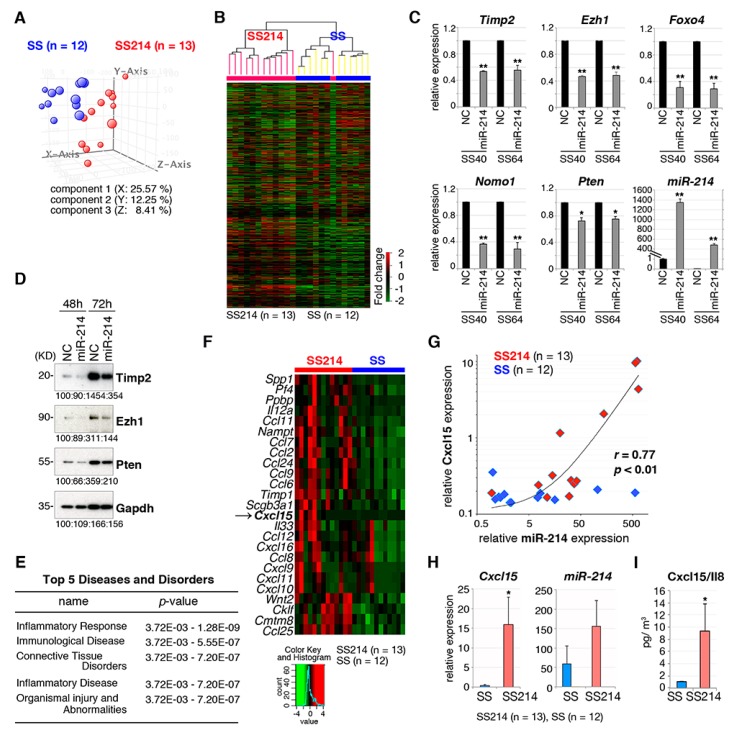
*miR-214* expression modulates gene expression in mouse synovial sarcoma. (**A**) Principal component analysis and (**B**) Hierarchical clustering of gene expression profiles of synovial sarcoma with and without *miR-214* overexpression. (**C**) RT-qPCR for candidate *miR-214* target genes, *Timp2*, *Ezh1*, *Foxo4*, *Nomo1*, and *Pten* in synovial sarcoma cells with or without *miR-214*. * *p* < 0.05, ** *p* < 0.01. (**D**) Immunoblotting for Timp2, Ezh1, and PTEN in the SS40 mouse synovial sarcoma cell with or without *miR-214* introduction. (**E**) Ingenuity Pathway Analysis (IPA) of signaling pathways for 272 upregulated genes in synovial sarcoma expressing *miR-214*. The key diseases and disorders are listed according to their ranking scores. (**F**) A heatmap of top 25 cytokine genes upregulated in synovial sarcoma with *miR-214* expression. The arrow indicates Cxcl15. (**G**) Pearson’s correlation analysis shows positive correlation between *miR-214* and *Cxcl15* expression in mouse synovial sarcomas. (**H**) Quantitative RT-PCR showed *Cxcl15* upregulation in *miR214*-positive synovial sarcomas. Expression of *miR-214* is shown in the right panel. *Cxcl15* expression relative to *Gapdh* and *miR-214* relative to *sno202* are shown. (**I**) IL-8 protein expression is compared between mouse synovial sarcomas with and without *miR-214*.

**Figure 6 cancers-12-00324-f006:**
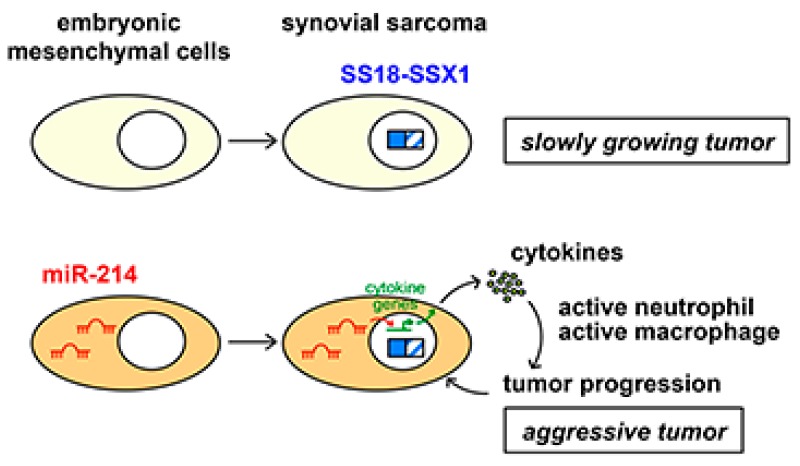
Proposed model of cooperation between *SS18-SSX1* and *miR-214*. *SS18-SSX1* expression in embryonic mesenchymal cells that do not express *miR-214* induces slowly growing tumors (**top**). On the other hand, *miR-214* upregulates expression of cytokine genes, such as *Cxcl15*, that activate neutrophils and macrophages in the tumor microenvironment, resulting in an aggressive nature of synovial sarcoma (**bottom**).

**Table 1 cancers-12-00324-t001:** Common retroviral integration sites in murine synovial sarcoma.

Chromosome Locus	Loci	No. of Integrations (*n* = 30)
1qH2.1	*Dnm3*	3
2qH3	*Tshz2*	2
2qB	*Cstad*	
4qD2.2	*Stk40*	
4qD3	*Hspg2*	
6qB3	*Creb5*	
8qA2	*Fgfr1*	
8qE2	*Irf2bp2*	
11qD	*Col1a1*	
12qF1	*Trmt61a*	
13qA4	*Nedd9*	
15qF3	*Itga5*	
17qE2	*Ltbp1*	
17qE4	*4933433H22Rik*	
18qE3	*Zbtb7c*	

**Table 2 cancers-12-00324-t002:** Upregulated cytokine genes in mouse synovial sarcoma *.

Genes	Fold Change	Products
***Cxcl15***	12.34	chemokine (C-X-C motif) ligand 15
*Ppbp*	5.80	pro-platelet basic protein
*Spp1*	4.31	secreted phosphoprotein 1
*Ccl6*	3.63	chemokine (C-C motif) ligand 6
*Ccl9*	3.57	chemokine (C-C motif) ligand 9
*Ccl11*	2.48	chemokine (C-C motif) ligand 11
*Cxcl9*	2.21	chemokine (C-X-C motif) ligand 9
*Ccl2*	2.16	chemokine (C-C motif) ligand 2
*Pf4*	2.13	platelet factor 4
*Ccl7*	1.88	chemokine (C-C motif) ligand 7

* Upregulated genes in *SS18-SSX1-* and *miR-214*-coexpressing tumors (versus *SS18-SSX1* only tumors). Cxcl15 that is upregulated both in mouse and human synovial sarcoma is indicated in bold.

**Table 3 cancers-12-00324-t003:** Upregulated cytokine genes in human synovial sarcoma *.

Genes	Fold Change	Products
*BMP7*	6.79	bone morphogenetic protein 7
*BMP5*	4.48	bone morphogenetic protein 5
*WNT5A*	4.35	wingless-type MMTV integration site family, member 5A
*CXCL12*	3.35	chemokine (C-X-C motif) ligand 12
*TNFRSF19*	2.75	tumor necrosis factor receptor superfamily, member 19
*BMP2*	2.46	bone morphogenetic protein 2
*TGFB2*	2.14	transforming growth factor beta 2
***CXCL8***	1.86	chemokine (C-X-C motif) ligand 8
*FAM19A1*	1.63	family with sequence similarity 19 (chemokine (C-C motif)-like), member A1
*CCL28*	1.60	chemokine (C-C motif) ligand 28

* Upregulated genes in human synovial sarcoma versus other sarcomas (osteosarcoma, Ewing sarcoma, alveolar soft part sarcoma, and rhabdomyosarcoma). CXCL8 that is upregulated both in mouse and human synovial sarcoma is indicated in bold.

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
