# Peer review of "Cooperation between SS18-SSX1 and miR-214 in Synovial Sarcoma Development and Progression"

_cancers, 2020, doi:10.3390/cancers12020324_

Round 1

Reviewer 1 Report

In the present paper Tanaka et al., have established an ex vivo mouse model of synovial sarcoma using retroviral-mediated gene transfer of the fusion protein SS18-16 SSX1 into mouse embryonic mesenchymal cells followed by subcutaneous transplantation into nude mice.

Through this approach the authors are able to produce a cell model of murine synovial sarcoma similar to the human tumor useful for preclinical studies. The work is clearly written, well presented and experiments are consequential.

The produced model of SS is in line with the known molecular characteristics of the tumor cells such as Tle1 and Atf2 up-regulation and Egr1 down-regulation. In an effort to identify drivers co-working with the fusion protein, the authors demonstrate that the retroviral insertion at the Dnm3os locus induces the up-regulation of two miRNAs, miR-214 and miR-199a2, and the first is the most important for the tumor phenotype. Indeed, when they co-express the fusion protein together with miR.214, the latency of tumor development in vivo is significantly reduced. However, they are unable to find any effect of the forced up-regulated miRNA on tumorigenic properties of MeSC cells and say that miR-214 functions in cell non- autonomous manner, promoting by cytokine gene expression a pro-tumor effect on microenvironment.

However, authors could try to verify by IHC whether the tumors expressing miR-214 have a higher number of CD31 vessels, due that IL8 has been related to neo-angiogenesis.

I also think that, to demonstrate the cooperative role of the miRNA in SS18-SSX1-induced SS, its expression should be counteracted with antimiRs in cells obtained from tumors expressing the miRNA (with integration) such as SS18, 19 or 23 to define their ability to grow in vivo and the expression of the identified cytokines. Authors can also overexpress miR-214 in tumor cells in which it is not so much up-regulated due to the absence of integration at the Dnm3 site, such as SS29 and follow the in vivo growth. However, since these in vivo experiments are really time-expensive due to the latency of the tumor cell growth, authors should at least perform these experiments in vitro and show the modulation of expression of the cytokines. An additional experiment in vitro could be to treat endothelial cells with the conditioned medium of the tumor cells in which miR-214 is counteracted or not and verify the capability of these cells to differentiate in vessels in vitro or to migrate in a wound healing assay. It is really simple to do.

Similarly, miR-214 antimiRs should be also applied in at least one human cell line of SS18-SSX-positive synovial sarcoma that authors used in Namatame et al., Oncotarget 2018, and the modulation of the expression of the same cytokines verified.

Major

Could the authors comment on the miR-214-induced up-regulation of the cytokines since usually miRNAs over-expression lead to down-regulation of direct targets? Have the authors verified the expression of pri-miR-214 and 199a? Just to demonstrate that the overexpression is transcriptionally induced and not a result of enhanced maturation. miR-199a2 did not seem to be up-regulated by retroviral insertion since only 2 out 6 samples showed high expression and one of these did not express the fusion protein. For both miRNAs there are some samples w/o insertion for which the expression was up-regulated also compared to some other with insertion. Can the authors comment on this? In my opinion, the data on either Fgfr1 or Nedd9 should be reported in supplementary data.

Minor

Is the time point for subcutaneous transplantation of 48h as reported for ASPS paper on Cancer Res 2016? Or is it different? This should be indicated also because it is important to explain the difference in gene expression between tumors and cells. It seems that the symbol for beta-catenin lacks as for lines 42 and 106 line 44 please correct “…the frequency of secondary mutations in this disease is not very high..” 2E should be better explained highlighting the common genes between cells and tumors too Methods and number of patients in the cohort used for Kaplan-Meyer curve lacks. Similarly, the number of primary samples for the analysis of expression in Fig- S2 should be reported for each tumor type. References about publicly available datasets should be reported. There are some typos errors also in the Figure legends (Fig S2: tisseus instead tissues) in line 258 maybe miR-199a2 should be miR-214 I missed Figure S6 Comments should be moved from Results to Discussion section

Reviewer 2 Report

The manuscript” Cooperation between SS18-SSX1 and miR-214 in synovial sarcoma development and progression” By Tanaka et al presents a highly interesting new mouse model for the enigmatic synovial sarcoma. The model recapitulates many features of this tumor type. Some of the data adhere to and recapitulate results from earlier studies but the new findings on miR-214 in vivo role for tumor growth and the suggested mechanism of action is a major progress in the understanding of synovial sarcoma.

The figures are nice and easy to understand. However, the figures   2 A and C presenting the GSEA  data cannot be clearly viewed and read as presented for this referee. Thee resolution or text size has to be improved before publishing.

Reviewer 3 Report

In the manuscript entitled „Cooperation between SS18-SSX1 and miR214 in synovial sarcoma development and progression” Tanaka et al describe a new ex vivo mouse model of synovial sarcoma. Using this model they are able to study the role of miR214 in sarcoma development, showing that high levels of miR214 cooperate with SS18-SSX1 in tumorigenesis. The authors further show that miR214 promotes cytokine gene expression providing a possible mechanism for miR214 involvement in synovial sarcoma progression.

 The study is relevant, well designed and the manuscript well written. Although a mouse model for synovial sarcoma has already been developed, the model presented here is more flexible allowing to quickly test cooperation with other co-occurring alterations. Additionally, while miR214 has been previously reported as highly expressed in human synovial sarcoma a role in SS progression has never been demonstrated.

Some points that should be addressed are:

The material and methods are short. More information on how experiments were performed should be provided. Ex: How long after eMCs were transduced are they implanted in nude mice? How many cells are injected? This are all important points that need to be clear if other researchers are interested in using this model. Ex2: Microarray analysis. What was used as normal tissue for comparison with SS murine tumors is unclear. How many tumors were analyzed and whether replicates were used for expression analysis in eMCs is also not mentioned.

The authors report that n=59 mice and penetrance was 100%. However, in the results section mentions that: “ Of the 33 sarcomas 27 were monophasic and 6 were biphasic”. Were only 33 tumors histologically analyzed? Or not all the 59 mice had tumors?

While the authors provide a characterization of murine tumors, it is unclear how many tumors were analyzed for SS markers (eg BCL2 and TLE1). Do all murine tumors consistently express high levels of these markers? Not necessary all tumors need to be analyzed but this information should be present. Ex: “From the X tumors analyzed X exhibited high immunoreactivity for Y and Z”.

Gene set enrichment analysis in Figure 2. A table with top GSEA enrichment scores and gene sets from the GSEA database should be presented in supplementary figures/table. The ones presented are indeed interesting but a more unbiased analysis should be shown.

Additionally, it is unclear if the gene sets used are from the GSEA database or were gene sets created by the authors. “SNF5 pathway” does not really mean much as there are no canonical genes regulated by SNF5. In the GSEA database the gene sets for SNF5 found are:

V1_UP. Genes up-regulated in MEF cells (embryonic fibroblasts) with knockout of SNF5 gene. V1_DN. Genes down-regulated in MEF cells (embryonic fibroblasts) with knockout of SNF5 gene. BIOCARTA_HSWI_SNF_PATHWAY (only 12 genes)

Please provide Gene set names (ref IDs) or information on how they were created.

Supplementary information. Table S1 only shows upregulated genes in response to SS18-SSX expression in ECMs or in tumors. All differentially expressed genes should be shown. Would also be important to show a list of differentially expressed genes in tumors overexpressing miR124 (data used for IPA analysis in Figure 5E).

In the paragraph, (line 247-249). “Although our current model did not contribute to the topological information for the cell-of-origin, results suggest an association between some developmental abnormalities and synovial sarcomagenesis.” What results the authors are referring to here? Their own or from others? No reference is provided.

A recent paper that described SS18-SSX1 target genes and dependence on polycomb complex 1 was not referenced in the introduction when describing known mechanisms of SS18-SSX1 activity (PMID:29502955).

The reason for mir214 upregulation due to integration at the Dnm3 locus in unclear and not discussed.
